Swedish high-school pupils’ attitudes towards drugs in relation to drug usage, impulsiveness and other risk factors

Mousavi Fariba 1 fariba_mousavi@yahoo.se
Garcia Danilo 1 2 danilo.garcia@neuro.gu.se danilo.garcia@euromail.se
Jimmefors Alexander 1
Archer Trevor 1 3
Ewalds-Kvist Béatrice 4 5
1 Network for Empowerment and Well-Being , Sweden
2 Institute of Neuroscience and Physiology, Centre for Ethics, Law and Mental Health, University of Gothenburg , Gothenburg , Sweden
3 Department of Psychology, University of Gothenburg , Gothenburg , Sweden
4 Department of Psychology, University of Turku , Finland
5 Department of Psychology, Stockholm University , Stockholm , Sweden
Patton Robert
Electronic publication date: 2014 Jun 5
Publication date: 2014
Volume: 2
Electronic Location ID: e410
Received 2014 Feb 19; Accepted 2014 May 12
Copyright: © 2014 Mousavi et al.
Copyright year: 2014
Copyright holder: Mousavi et al.
License: This is an open access article distributed under the terms of the Creative Commons Attribution License, which permits unrestricted use, distribution, reproduction and adaptation in any medium and for any purpose provided that it is properly attributed. For attribution, the original author(s), title, publication source (PeerJ) and either DOI or URL of the article must be cited.
License URL: https://creativecommons.org/licenses/by/4.0/

Keywords: Attitudes towards drug use, Impulsiveness, Gender and drug use, Risk factor for drug use

Funding: Stiftelsens Kempe-Carlgrenska Fonden This research was supported by the Stiftelsens Kempe-Carlgrenska Fonden. The funders had no role in study design, data collection and analysis, decision to publish, or preparation of the manuscript.

==============================
Background. Illicit drug use influences people’s lives and elicits unwanted behaviour. Current research shows that there is an increase in young people’s drug use in Sweden. The aim was to investigate Swedish high-school pupils’ attitudes, impulsiveness and gender differences linked to drug use. Risk and protective factors relative to drug use were also a focus of interest.

Method. High school pupils (n = 146) aged 17–21 years, responded to the Adolescent Health and Development Inventory, Barratt Impulsiveness Scale and Knowledge, and the Attitudes and Beliefs. Direct logistic, multiple regression analyses, and Multivariate Analysis of Variance were used to analyze the data.

Results. Positive Attitudes towards drugs were predicted by risk factors (odds ratio = 37.31) and gender (odds ratio = .32). Risk factors (odds ratio = 46.89), positive attitudes towards drugs (odds ratio = 4.63), and impulsiveness (odds ratio = 1.11) predicted drug usage. Risk factors dimensions Family, Friends and Individual Characteristic were positively related to impulsiveness among drug users. Moreover, although boys reported using drugs to a greater extent, girls expressed more positive attitude towards drugs and even reported more impulsiveness than boys.

Conclusion. This study reinforces the notion that research ought to focus on gender differences relative to pro-drug attitudes along with testing for differences in the predictors of girls’ and boys’ delinquency and impulsiveness. Positive attitudes towards drugs among adolescents seem to be part of a vicious circle including risk factors, such as friendly drug environments (e.g., friends who use drugs) and unsupportive family environments, individual characteristics, and impulsiveness.

Introduction

A drug abuser may be defined as an individual who has relinquished control over his/her life to psychoactive substances (Fraser & Moore, 2008). This condition produces altered neurological functions, and changed perceptions, moods, consciousness and energy levels (King, 2008). The user turns into an ‘abuser’ when a drug impacts his/her normal functioning and well-being (Johnston & O’Malley, 2001). The label ‘abuser’ covers the inappropriate use of any substance, especially those that alter consciousness (e.g., alcohol, cocaine methamphetamines) and generate significant distress and function impairment (Medical Dictionary, 2013). Drug abuse, although disapproved by most societies, may involve the illegal use of drugs for recreational purposes or for the relief of medical problems without a health care practitioner’s recommendation (Merck Manuals, 2009). Drugs have been used universally in all cultures and all social classes, for example alcohol was a common intoxicant in ancient Greece, in South America, Indians have chewed the leaves of the raw material that make up cocaine production (Goldberg, 1993), and opium has been used as both an intoxicant and medicine in many cultures, especially China (Ramström, 1983). Several studies have contributed to understanding of the role of drugs in different cultural contexts, such as drinking contributes to British identity among Australian skinheads (Moore, 1994) and the pub culture can be understood as an expression of the working class and ‘masculinity’ (Sulkunen et al., 1997). The global annual prevalence of illicit drug users was estimated to be 3.30–6.10% in people aged 15–64 years in 2009 (United Nations Office on Drugs and Crime, 2011). Cannabis is the most frequently used drug with a projected global annual prevalence rate of 2.80–4.50%, 10.70% in North America and 6.80% in Europe (European Monitoring Centre for Drugs and Drug Addiction, 2010a; European Monitoring Centre for Drugs and Drug Addiction, 2010b).

In this context, according to alcohol sales statistics, beer intake is dominant in most of Europe, including Scandinavia, whereas wine is consumed more often in Southern Europe. In Sweden for instance, the average consumption for an adult is assessed to be 8.85 liters of alcohol per year (Andersson, Moller & Galea, 2012), while 2.30% of 16–84 year-old individuals use cannabis for recreational purposes (National Institute of Public Health [NIPH], 2011). Among adolescents, 20% of Swedish boys and 15% of Swedish girls have used drugs at one time or another. In upper secondary grade school, beer is dominant among Swedish boys (48%), followed by spirits (25%), and mixed drinks (13%). Among Swedish girls, mixed drinks (32%) and spirits (27%) are the most consumed forms of alcohol (Henriksson & Leifman, 2012). According to the NIPH (2009/10), Swedish school children’s drug usage has increased slightly in recent years. Drug availability is considerably enhanced and is linked to positive attitudes to experiment with alcohol and other drugs. While the proportion of drug usage seems to have levelled off among girls, it has risen from 16–17% in 2004–2008 to 21% in 2010 among boys (Henriksson & Leifman, 2012). For instance, Taylor Nelson Sofres Sifo (TNS Sifo, 2012) surveyed all high-school students in Stockholm, including the high school targeted in the present study. The TNS Sifo Gallup survey showed that 27% of the boys and 15% of the girls had tried drugs during the year 2012. The high school included in the present study had a drug-user increase from 16% to 21%. This observation implies that the number of adolescents expressing a more positive attitude towards drugs may have increased.

Attitudes towards drugs

An attitude is defined as a psychological tendency expressed by an approval or a disapproval of a person or thing. In particular, attitudes facilitate an individual’s judgment for goal achievements, determination of consequences or conveyance of attitudes to other individuals and an individual seeks an environment with attitudes consistent with his own (Augustusson, 2005). Changes in an attitude may be perceived as an attempt to balance the social environment (Helkama, Myllyniemi & Liebkind, 2004), for example, peer conformity (Aronson, Wilson & Akert, 2005). Nevertheless, Rytterbro (2006) and Rödner and colleagues (2007) revealed an ongoing general liberalization of attitudes towards drugs among young people. Drug users seem to attribute positive effects to illicit drugs to ‘play down’ their negative effects—for example, by believing that cannabis is less harmful than other drugs and that cannabis use may not be perhaps as harmful as alcohol.

Parental knowledge concerning teenage activity and residence are also important predictors of drug abuse. However, it is not the parents’ active questioning or monitoring per se, but the teenager’s own narrative that constitutes an important basis for our understanding on drug usage (Kakihara et al., 2010; Keijsers et al., 2010; Kerr, Stattin & Burk, 2010). The likelihood that a young person develops into a drug user and abuser is increased through peer pressure, at an age when familiarity with negative abuse effects is limited. Furthermore, young people have difficulties determining if friends are using drugs or not (Anderson, 1991). Moreover, compared to girls, boys are exposed earlier to intoxicating substances (Van Etten & Anthony, 2001) and have a greater liability for lifetime prevalence of exposure to illicit substances (Aarnoudse, Dieleman & Stricker, 2007; Gray, 2007). It appears that the pattern in female drug usage is related to some extent to intimate relationships, while the male model links to independence and freedom (Trulsson, 2006).

Risk factors and protective aspects relative to drug abuse

Risk factors for drug use comprise uninvolved parents, peer pressure, hostility towards the child and harsh punishments, poor school or academic achievements, low socioeconomic status and availability of drugs (Hawkins, Catalano & Miller, 1992; Merline et al., 2004; Schuster et al., 2001). Additionally, attention deficit and hyperactivity disorder, personality traits such as lack of empathy (i.e., low communal values or cooperativeness), impulsiveness (i.e., low agency or self-directedness), non-attendance in local environmental issues (i.e., low self-transcendence), fearlessness, sensation-seeking and lack of emotion regulation constitute individual-specific risk factors for drug abuse (Andershed & Andershed, 2005; Loeber & Farrington, 1998). Importantly, a personality characteristic such as impulsiveness is a major contributor to drug consumption and having a positive attitude towards drugs, which in turn also increases risky behaviour (Hawkins, Catalano & Miller, 1992). At the personal level, a human being’s level of vulnerability constitutes an individual-specific risk factor, which puts the person at danger for developing antisocial and aggressive behavior (Gross, 2007). Poor emotional regulation leads to, instead of using cognitive strategies, the use of physical violence to retaliate, especially among males (Kåver & Nilsonne, 2002). Women, for instance, are known to use on average less drugs than men (Van Etten & Anthony, 2001). Kloos and colleagues (2009), for instance, suggested social and cultural norms might explain gender differences in drug abuse. Traditionally, females fear to lose control in a social context; consequently fewer women succumb to drug misuse whereas drug consumption may serve a purpose in regulating emotions, especially anger and impulsiveness (Kloos et al., 2009).

Conversely, health-related behaviour in adolescence is influenced by immediate social and environmental factors, such as closeness, cohesion and care of family, which lower the risk for substance abuse (Duncan, Duncan & Strycker, 2003; Hill et al., 2005; Pires & Jenkins, 2007; Sale et al., 2005). Stattin & Kerr (2000) found that parents with rules for their teens decreased the risks for antisocial behaviour (see also Kakihara et al., 2010). On the other hand, parents who communicate with their teens convey a better understanding by supporting and guiding them. Teenagers who have a good and respectful relationship with their parents are more likely to imitate their parents’ attitudes, which may affect their use of alcohol and drugs (Keijsers et al., 2010; McNeely & Barber, 2010). Close relationships promote transparency and reduce the risk that the teenager would engage in antisocial behaviour (Vieno et al., 2009).

The present study

The purpose of the present study was to investigate high-school pupils’ attitudes towards drugs, impulsiveness and other risk factors relative to their use of drugs for non-medical reasons. Due to the widespread and complex aspect of the problem, only three specific research questions were examined in the present analysis:

1. Which factors contribute to high-school students’ positive attitude towards drug usage?

2. Which factors contribute to high-school students’ drug usage?

3. Which factors contribute to drug users’ impulsiveness?

Method

Ethical statement

After consulting with the university’s Ethical Review Board (University of Gothenburg) and according to law (2003: 460, section 2) concerning the ethical research involving humans we arrived at the conclusion that the design of the present study (e.g., all participants’ data were anonymous and will not be used for commercial or other non-scientific purposes) required only informed consent from participants and a signed consent from the principal of the participating high school.

Participants and procedure

Altogether 15 high-schools’ principals in Stockholm, Sweden, were approached until a principal for a high school agreed to participate in the present study. The staff of the schools that declined to participate did so due lack of time or found the drug issue to be irrelevant to students’ curriculum. At the participating high school, a total of 160 questionnaires were handed out to the pupils aged 17–21. In this part of Sweden (the Stockholm region), drug issues are a known problem (CAN, 2012; NIPH, 2009/10 & TNS Sifo, 2012). Fourteen (9%) pupils refused to participate or did not complete the forms accurately and were thus excluded from the study. Accordingly, the sample comprised 146 (91%) pupils who attended a 3-year Natural Science or Social Science program. The boys (47.30%) were on average 18.20 (SD = 0.65) and the girls (51.40%) were on average 18.03 (SD = 0.57) years. Their parents had educational levels ranging from: no education (1.40%), high school (8.90%), upper secondary school (17.10%), vocational education (1.40%) to university (52.60%). This suggests that a majority of the pupils had parents with higher education. A total of 15.10%, however, did not respond to this question. The majority of the participants were Swedish (n = 143), 1 was from Russia, 1 was from Georgia and 1 was from Iran. The majority of the pupils (88.40%) indicated a big city as their place of upbringing, 10.30% indicated a small city and 1.40% did not answer this question.

The survey was conducted at the school during an English lecture at high-school C level. The researcher delivered the questionnaire to the school principal. The questionnaire (127 questions) comprised measures of impulsiveness, attitudes towards drugs, protective and risk factors for students’ drug use, and some background variables. Furthermore, before handing out the questionnaires the researcher received a written assent letter signed by the principal. Then in turn, the principal informed every C-level English teacher that they would ensure that the students participated in the survey and completed the questionnaires during the English lesson. The pupils were informed that the study was anonymous, voluntary, required a duration of 45 (±5) min, and that they were free to discontinue the completion of the form whenever they wanted without any justification. After completion, pupils were instructed to seal the survey in an envelope that was handed to the teacher. Data collection took place from mid-November 2012 to January 2013.

Statistical treatment

By means of linear and logistic regression analyses as well as Multivariate Analysis of Variance (MANOVA) students’ use or non-use of drugs and attitudes towards drugs provided the dependent variable, while gender, age, level of impulsiveness, risk and protective factors constituted the independent variables. To avoid a too small sample, 146 questionnaires (×127 questions) were collected which well exceeded the requirement of at least 15 individuals per predictor in regression analysis (Pallant, 2001). This sample size also reduced the occurrence of false significances in MANOVA.

Measures

Participants’ background

The background instrument comprised 5 items about socio-demographic data including the respondent’s age, gender, home country, place of upbringing and level of parents’ education.

Drug use

This part of the form contained a total of 4 items. Participants were asked to indicate if they have used drugs for non-medical reason (Yes, No) the type of drugs the respondent had used, his/her age at the first use of various drugs and the frequency of drug use.

Attitudes towards drugs

The Knowledge, Attitudes and Beliefs inventory (Bryan et al., 2000), was modified for this study and consisted of 21 items in which participants answer the questions regarding their attitudes to drug use (e.g., “Our society is too tolerant towards drug users”, “Occasional use of cannabis is not really dangerous”, “It is normal that young people will try drugs at least once”, “Reports about the extent of drug usage amongst young people are exaggerated by the media”). The items were answered using a 7-point Likert scale (1 = Disagree strongly. 2 = Don’t agree. 3 = Agree strongly. 4 = Agree moderately. 5 = Agree slightly. 6 = Don’t know. 7 = I don’t care). For the purpose of the present study, and as recommended by Bryan and colleagues, the response options were collapsed into two categories (Agree and Disagree). In other words, categorizing participants in those who had a positive attitude towards drugs and those who did not had a positive attitude towards drugs. Nevertheless, using the whole scale the Cronbach’s alpha coefficient for this measure was .72.

Risk and protective factors

The Adolescent Health and Development (Jessor, Turbin & Costa, 1998b) and the Communities That Care (Hawkins, Catalano & Miller, 1992) questionnaires assess a variety of behaviours as well as a range of risk and protective factors in different domains (3–4 items for each domain) using a 4-point Likert scale (1 = Almost always, 4 = Almost never). The domains include Family (Risk factor item example: “People in my family often insult or yell at each other”; Protective factor item example: “My parents give me lots of chances to do fun things with them”), Community (Risk factor item example: “I would like to get out of my neighbourhood”; Protective factor item example: “There are people in my neighborhood who encourage me to do my best”), Friends (Risk factor item example: “How wrong do you think it is for someone your age to smoke marijuana?”; Protective factor item example: “If you were doing something that is bad for your health, would your friends try to get you to stop?”), and Individual Characteristics (Risk factor item example: “I do the opposite of what people tell me, just to get them mad”; Protective factor item example: “It is important to be honest with your parents, even if they become upset or you get punished”). We also constructed a total score for measuring risk and protective factors as a whole, by simply adding all items in the risk and protective domain. The reliability by Cronbach’s alpha for risk factors with 32 items was .83 and for protective factors with 14 items .84.

Impulsiveness

The Barratt Impulsiveness Scale, (BIS-11; Patton, Stanford & Barratt, 1995) contains a total of 30 items, each of which is answered on a 4-point Likert scale (1 = Rarely/never. 4 = Almost always/always). The level of impulsiveness is calculated by summing up the scores for each item, the higher score, the more impulsiveness. The Cronbach’s alpha for 29 items, after factor analysis, was .84.

Results

Respondent characteristics as well as the means and standard deviations for different measurements performed are provided in the supplemental information (Table S1). An explorative analysis, before testing the specific research questions, showed that boys used more drugs (41% of the boys compared to 21% of the girls). A total of 4.80% of the pupils who indicated using drugs reported using alcohol, 5.50% of these pupils reported using cannabis, 4.80% marijuana and 16.40% did not answer this specific question. The frequency of drug usage was 0.70% weekly, 0.70% monthly, 3.40% only once, 0.70% every two months, 0.70% every three weeks, 1.40% just three times, 0.70% sometimes not very often, 1.40% only two times, and 21.90% did not answer. The distribution of pupils reported when they started using drugs was as follows: 2.70% by age 13, 7.50% by age 15, 8.90% by age 16, 8.20% by age 17, 2.70% by age 18, and 1.40% did not answer the question.

Chi-square for independence testing was used to explore the relationship between gender and positive attitudes towards drugs and gender and drugs usage. A Chi-square test for independence (with Yates Continuity Correction) indicated significant association between gender and positive attitudes towards drugs, X2 (1, n = 118) = 10.89, p = .001, phi = .32. All expected cell sizes were greater than 5 (in this case, greater than 23.67). Even a Chi-square test for independence (with Yates Continuity Correction) indicated significant association between gender and drugs usage, X2 (1, n = 144) = 5.40, p = .02, phi = .21. Also here, all expected cell sizes were greater than 5 (in this case, greater than 21.08). The phi coefficients in both analyses (.32 for positive attitudes towards drugs and .21 for drugs usage) can be considered a medium and small effect size, respectively, using Cohen’s (1988) criteria.

An independent-samples t-test was conducted to compare the total sum of risk factors scores for males and females. There was a significant difference (t (121) = 1.95, p = .053, two-tailed) in scores for males (M = 67.61, SD = 10.77) and females (M = 63.95, SD = 9.97). The magnitude of the differences in the means was small (eta squared = .03). For complementary analyses, the numbers of included items as well as the value of Cronbach’s alpha for each instrument, see Table S1.

Attitude towards drugs

Direct logistic regression was performed to assess the impact of factors on the likelihood that respondents would report that they were likely to express a positive attitude towards drugs. The model contained five factors or independent variables (gender, age, impulsiveness, total sum of risk factors and total sum of protective factors). The full model containing all predictors was statistically significant (χ2 (5, N = 117) = 30.27 p < .0001). That is, the model distinguished between respondents who were categorized as having a positive attitude towards drugs from those who were categorized as not having a positive attitude towards drugs. The model as a whole explained between 22.80% (Cox and Snell R square) and 30.70% (Nagelkerke R squared) of the variance in attitudes towards drugs, and classified correctly 76.10% of these cases. As shown in Table 1, two of the independent variables made a unique statistically significant contribution to the model, gender and the total sum of risk factors. The strongest predictor of reporting positive attitudes towards drugs was the total sum of risk factors with an odds ratio of 37.31. This indicated that respondents who live in more risk factor-prone environments (including Family, Community, Friends, and Individual Characteristics) were over 37 times more likely to report a positive attitude towards drugs than those who did not live under such risk factors, controlling for all other variables in the model.

Table 1 Logistic regression analysis predicting respondents’ attitude towards drugs.

	B	S.E.	Wald	df	Sig.	Exp(B)	95% CI for EXP(B)	
							Lower	Upper	
Gender	−1.15	0.44	6.84	1	0.009	0.32	0.13	0.75	
Age	0.27	0.33	0.65	1	0.419	1.31	0.68	2.52	
Impulsiveness	−0.00	0.03	0.03	1	0.872	0.99	0.95	1.05	
Risk factors	3.62	0.98	13.54	1	0.000	37.31	5.43	256.43	
Protective factors	−1.38	0.75	3.37	1	0.066	0.25	0.06	1.10	
Constant	−9.27	6.07	2.33	1	0.127	0.00			

Drug usage

Direct logistic regression was performed to assess the impact of factors on the likelihood that the respondents would report that they had used drugs. The model contained 6 independent variables (age, gender, totals sums of risk factors, total sum of protective factors, impulsiveness, and attitudes towards drugs). The model containing all predictors was significant (X2 (6, N = 117) = 49.41, p = 0.0001), thereby indicating that the model was able to distinguish between respondents who reported using drugs and those who reported not using drugs. The model as a whole explained between 34.4% (Cox and Snell R square) and 48.1% (Nagelkerke R squared) of the variance in drug use and correctly classified 84.6% of the cases. As shown in Table 2, three of the independents variables made an unique contribution to the model (totals sums of risk factors, impulsiveness and attitudes towards drugs). The strongest predictor of drug usage was risk factors (including Family, Community, Friends, and Individual Characteristics), recording an odds ratio of 46.89. This indicates respondents with high-risk factor were over 46 times more likely to use drugs (see Table 2 for the details). Attitudes towards drugs also predicted the likelihood of being a drug user with an odds ratio of 4.63, even more so than impulsiveness with an odds ratio of 1.11.

Table 2 Logistic regression analysis for likelihood that the respondents would report that they had used drugs.

Variable	B	S.E.	Wald	df	Sig.	Exp(B)	95% CI for EXP(B)	
							Lower	Upper	
Gender	−0.38	0.55	0.47	1	0.495	0.69	0.23	2.03	
Age	−0.21	0.41	0.27	1	0.604	0.81	0.37	1.79	
Risk factors	3.85	1.26	9.28	1	0.002	46.89	3.94	557.95	
Protective factors	−1.63	0.95	2.95	1	0.086	0.20	0.03	1.26	
Impulsiveness	0.11	0.03	10.08	1	0.002	1.11	1.04	1.19	
Atittudes towards drugs	1.53	0.55	7.68	1	0.006	4.63	1.57	13.68	
Constant	−9.19	7.43	1.53	1	0.216	0.00			

Risk factors contributing to drug users’ impulsiveness

A MANOVA was performed to investigate impact of age groups, gender and drug use as independent variables. Protective factors were not found significant in the analyses above, thus, not included in the MANOVA. In contrast, the risk factors were significant and therefore in order to further disentangle which risk factors contributed to drug users’ impulsiveness, we used each domain as independent variables: Family, Community, Friends, and Individual Characteristics. Preliminary assumption testing was conducted to check for normality, linearity, univariate and multivariate outliers, homogeneity of variance–covariance matrices, and multicollinearity, with no serious violations noted. There was a statistically significant difference between drug users and non-users on the combined dependent variables (F (4, 116) = 7.14, p = 0.0001; Wilks’ Lambda = 0.80; partial eta squared = 0.19). When the results for the dependent variables were considered separately, the only difference to reach statistical significance, using a Bonferroni adjusted alpha level of 0.02, were the risk factor domains of Family (F (1, 119) = 8.10, p = 0.005, partial eta squared = 0.06), Friends (F (1, 119) = 16.38, p = 0.0001, partial eta squared = 0.12), and Individual Characteristics (F (1, 119) = 14.91, p = 0.0001, partial eta squared = 0.11). The Family risk factor domain had a significant impact on impulsiveness (F (1, 119) = 5.59, p = 0.02, partial eta squared = 0.05) for the interaction between age group and drug use, see Table 3 for the details.

Table 3 Family, community, friends and individual characteristics as risk factors for drug user’s impulsiveness as indicated by the MANOVA.

Source	Tests of between-
subjects effects	Type III sum
of squares	df	Mean
square	F	Sig.	Partial eta
squared	
	Dependent variable							
Age groups	Family	311.22	1	311.22	3.27	0.073	0.03	
Community	32.60	1	32.60	2.09	0.151	0.02	
Friends	0.00	1	0.002	0.00	0.965	0.00	
Individual characteristics	4.40	1	4.40	0.89	0.348	0.01	
Drug use	Family	771.42	1	771.42	8.10	0.005	0.06	
Community	18.48	1	18.48	1.18	0.279	0.01	
Friends	15.24	1	15.24	14.92	0.000	0.11	
Individual characteristics	81.04	1	81.04	16.38	0.000	0.12	
Age groups ∗
Drug use	Family	532.93	1	532.93	5.59	0.020	0.05	
Community	8.81	1	8.81	0.56	0.454	0.01	
Friends	0.15	1	0.15	0.15	0.700	0.00	
Individual characteristics	3.00	1	3.00	0.61	0.438	0.01	

In sum, individuals’ positive attitude towards drugs is impacted by the total sum of risk factors. Further, the total sum of risk factors, impulsiveness and attitudes towards drugs predicted drug usage. Then again, risk factors dimensions: family, friends and individual’s characteristics predicted impulsiveness among drug users.

Discussion

The purpose of this study was to investigate high-school students’ attitudes towards drugs, impulsiveness and other risk factors relative to their use of drugs for non-medical reasons in Stockholm, where drugs were known to be a problem (CAN, 2012; NIPH, 2009/10; TNS Sifo, 2012). It was observed that gender and the total sum of risk factor scores predicted positive attitudes toward drug use. The risk factors involve absentee parents, peer-group pressure, hostility towards the child and harsh punishments, poor school or academic achievements, low socioeconomic status as well as the availability of drugs. According to Augustusson (2005), attitudes are part of an existing general social discourse and, currently, young people spend more time outside their family and are more influenced by peers, friends and surroundings than by their own family (Kakihara et al., 2010; Keijsers et al., 2010; Kerr, Stattin & Burk, 2010; Stattin & Kerr, 2000; Vieno et al., 2009). Thus, the development of positive attitudes towards drugs appears to be a combination of risk factors allowing the exposure of the adolescents to a general social discourse. Indeed, teenagers seek out friends with similar interests and attitudes. In this context, social and cultural norms may elucidate gender differences in substance abuse (Kloos et al., 2009). Normally, young males, compared to young females, are often exposed earlier to illicit substances (Van Etten & Anthony, 2001). In addition, the present study not only revealed that more males ‘tried’ drugs, but also that more females maintain positive attitudes towards drugs. This observation may imply changes in attitudes in a desire to achieve a balance with the social environment (Helkama, Myllyniemi & Liebkind, 2004). In other words, girls might adjust to the “norm” out of fear for exclusion from their peer group (Aronson, Wilson & Akert, 2005). At the same time, some researchers suggest that gender differences in drug usage may be explained from an expected gender role perspective—women fear, more so than men, losing control in a social context (Kloos et al., 2009).

Attitudes towards drugs predicted drug usage. Together with the results suggesting that risk factors lead to positive attitudes towards drugs, our results reveal a ‘vicious circle’ leading to drug usage, which in turn might lead to further risk factors (e.g., exposure to drug environments). With regard to drug usage, as in most studies, impulsiveness was found also to be a predictor of drug usage. Additionally, friends and family constituted threats that contributed most to a teenager’s impulsiveness and drug use propensity, thereby implying individual vulnerability combined with a propensity for antisocial and aggressive behaviour (see Gross, 2007). Indeed, parental guidance combined with support and consequential relationship may prevent drug usage among teenagers (Keijsers et al., 2010; McNeely & Barber, 2010; Stattin & Kerr, 2000). Parental monitoring and attention facilitates caution in teenagers for choice of peer-association and involvement in risky activities (Vieno et al., 2009). Teenagers’ peers constitute risk factors when young people have difficulties in setting limits for themselves and find it difficult to distinguish between right from wrong (Anderson, 1991), that is, teenagers high in impulsiveness.

Limitations of the study

The findings from the current study were based on cross-sectional data; therefore, no causal direction may be specified. For example, do the expressions of impulsiveness imply risky behaviour or some alteration of reward circuits or an epigenetic predisposition? The sample may not be representative of schools across Sweden, or for that matter a region, despite the school being known for drug problems. For instance, 14 out of the total 15 high school principals that were approached about study participation declined to partake due to the nature of the survey. Additionally, self-assessments are subjective measures and may be affected by both personality traits and dishonest responses (Watson, Clark & Tellegen, 1988). Although the questionnaire was composed of 127 items, the data offer just a limited portion of information regarding substance use and substance use problems experienced by high-school students in Sweden. Self-reported drug use may have been restricted due to fears of discovery since the survey was completed during an English lecture. Nevertheless, the instruments used here are well-validated and reliable. Finally, the questionnaires were in English, which implies that all the statements retained their original meaning, but it might have distorted the answers. Nevertheless, the principal accepted participation especially because students in this school are well known for their good English.

Future research

An individual’s vulnerability for addiction is modulated through several domains including emotional, social, cognitive and a variety of genetic and epigenetic factors (Andershed & Andershed, 2005; King, 2008; Merline et al., 2004; Nestler, 2012; Schuster et al., 2001). Female high-school students expressed a positive attitude towards the ‘normality of drug use’ reflecting possibly a liberal outlook (Rytterbro, 2006; Rödner & Olsson, 2007). Future studies should focus on external generalization and long-term trends from samples to different populations. From a cultural perspective, the shared values, norms and ideals expressed about drugs can be understood in terms of the culture that speaks for a social marginalization where drug use in the youth culture described as a normalization trend (Sørensen, 2000). This study reinforces the notion that research ought to focus on gender differences relative to pro-drug attitudes along with testing for differences in the predictors of girls’ and boys’ delinquency and impulsiveness.

Conclusion

An increase in drug use among high-school students was reported with both family and friends as risk factors as well as individual factors, such as impulsiveness. Male students reported using more drugs, but female pupils expressed more positive pro-drug attitudes. Further, female pupils reported that they had increased their use of drugs compared to earlier findings (TNS Sifo, 2012). This observation was hypothesized to constitute a signal for a social change, defined as a change of the norms, values, cultural products and symbols of the society. The pupils’ conduct may be interpreted also as an attempt to fit into the “normal” peer group as well as an effort to achieve a balance between individual structures and the social environments. Parental involvement and close relationships promote transparency and reduce the risk that the teenager engages in antisocial behaviour. Importantly, positive attitudes towards drugs among adolescents seem to be part of a ‘vicious circle’ including risk factors, such as friendly drug environments (e.g., friends who use drugs), unsupportive family environments, individual characteristics, and impulsiveness. All of which contribute to the tendency for drug usage (see Fig. 1).

Figure 1 A vicious circle including positive attitude towards drugs, risk factors and impulsiveness.

All increase the risk of using drugs.

Supplemental Information

Table S1 Respondent’s characteristics

Click here for additional data file.

Additional Information and Declarations

Competing Interests

Author Contributions

The authors declare there are no competing interests.

Fariba Mousavi and Béatrice Ewalds-Kvist conceived and designed the experiments, performed the experiments, analyzed the data, wrote the paper, prepared figures and/or tables, reviewed drafts of the paper.

Danilo Garcia wrote the paper, prepared figures and/or tables, reviewed drafts of the paper.

Alexander Jimmefors and Trevor Archer wrote the paper, reviewed drafts of the paper.

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
