# Peer review of "Swedish high-school pupils’ attitudes towards drugs in relation to drug usage, impulsiveness and other risk factors"

_PeerJ, doi:10.7717/peerj.410_

## Round 0.1 · original submission · Major Revisions

Please find attached the reviews of your paper. Please pay particular attention to the concerns of reviewer 2 regarding ethics - if you are unable to demonstrate ethical clearance for the study you must provide justification as to why this was not required

·

Basic reporting

The uniqueness and contribution of the present study to the existing literature should be elaborated in the introduction. Author(s) should elaborate the introduction from cultural perspective to demonstrate the contribution of manuscript to the literature from international perspective. The author(s) need to clarify the uniqueness and the contribution of the study. What are the theoretical and more importantly practical implications of the study? Both implications should be discussed in both in the introduction and discussion

Experimental design

Although there are many strengths of the study (e.g., sample size, a variety of measures to establish), I have provided (what I hope are) constructive critiques (i.e., areas that need to be addressed) to make this worthy of publication.

Validity of the findings

I commend the author(s) on taking the time to conduct analyses on a measure to ensure that it has adequate reliability and validity in sample from a culture where this measure has not been used. Researchers frequently skip this important step prior to conducting their research.

Additional comments

Next, there were many spacing errors (e.g., no space between words, no space between punctuation and word, double spaces between words). On a related note, in APA formatting, there is a space on each side of the “equal sign” (p = .05) and “less than sign” (p < .01). There were also random punctuation errors (e.g., semi-colons after a citation before the parenthesis.

Next, there were some grammatical errors.
And finally, in the reference section, many of the references did not follow APA formatting.

Reviewer 2 ·

Basic reporting

- Introduction needs restructuring, should provide stronger focus on adolescent/young adult drug use (and associated developmental risks)
- Prevalence of use rates should be provided for adolescent/young adult population (e.g., no percentage provided for NIPH 2009/10 citation)
- TNS Sifo citation cannot be accessed (invalid link), acronym should be explained
- Cannabis use references (EMCDDA 2010, NIPH 2011) lack context; no further reference to primary drug of use in remaining manuscript, no comparison to consumption rates for alcohol or other drugs
- Significant grammar/sentence construction errors (e.g., abstract, lines 90, 205, 209-11, 224, 233-235, 261, 280)
- Punctuation mistakes (e.g., lines 11, 118-9, 145, 208)
- Typos (e.g., Table 3; lines 19, 275, 251)
- Inconsistent citation style (e.g., lines 70, 262, 280)

Experimental design

- Possible breach of ethical standard: ethics statement missing from Materials and Methods section; no information provided on whether informed consent was obtained from students (or parents in case of underage students). This seems particularly concerning as 14 out of the total 15 high school principals that were approached about study participation declined to partake.
- Questionable appropriateness of materials used: The authors provide no explanation as to why no validated Swedish-language instruments were used. Even though the questionnaires were administered during an English lecture, it remains uncertain whether all students had sufficient English language skills to understand all questionnaire items and answer accurately.
- Students had to hand questionnaires to their teachers (as opposed to independent researchers) → strong risk for underreporting of drug use / general bias in self-report data
- Measures - Drug use: Drug use (line 148) comprised 4 items (not 3 items as reported)
- Measures - BIS-11: What are the cut-off values for normal versus pathological impulsiveness? (reported cut-off values by Patton et al 1995 and Stanford et al 2009 are inconsistent)

Validity of the findings

- Gender group comparisons (lines 189-196): report t and p-values in Results section
- Why are risk factors pooled? What is contribution of individual risk factors? Which individual risk factors contribute significantly, which do not?
- Lines 249-51: impulsiveness not previously reported as predictor for positive attitudes
- Line 251: individual personality traits not previously reported
- Line 278-282: the potential reasons cited for the observed gender differences are speculative, NOT based on study results
- Tables: neither drug use (4 items) nor demographics (5 items) appropriately reported
- Line 322: missing citation
- Figure 1: current figure does not add value to manuscript, would profit from integration of study findings (e.g., regression weights)

Additional comments

'Major Revisions' recommendation based on possible breach of ethical standard - please provide evidence of ethics approval and informed consent procedure

---

## Round 0.2 · Minor Revisions

Thank you for your revised submission. The manuscript now requires minor revision prior to our acceptance. Please note that the reviewer suggests extensive proof reading to correct a number of grammatical errors in the paper

·

Basic reporting

- Large sections of introduction are lengthy and/or lack relevance (e.g., lines 15-23, 47-56, 84-90)
- Uniqueness and contribution of the present study to the existing literature not well explained
- Logical flaws: lines 65/66 vs "peer pressure" (lines 66/67 and 74); individual characteristics summarized as high-risk environment (lines 256/57)
- Terminology issues: "pure alcohol" (lines 34/35) is not legally available, cannabis vs marijuana (line 204)
- Significant grammar/punctuation/spelling mistakes throughout manuscript - will require thorough revision prior to resubmission
- References do not conform with recommended PeerJ style (see https://peerj.com/about/author-instructions/)

Experimental design

- No information provided on whether informed consent was obtained from parents in case of underage students
- Please specify name of the university whose ethical review board was contacted

Validity of the findings

- Results - Attitude towards drugs: model (line 230) comprised 5 variables (not 4 variables as reported)
- Results - Table S1 (line 201) not included?
- Figure 1: current figure is misleading, should be converted into path diagram or similar

Additional comments

Due to the abundance of language/basic reporting issues, I strongly recommend that at least two authors proofread (and revise) the manuscript.

---

## Round 0.3 · accepted · Accept

many thanks for the revised submission - I am satisfied that you have addressed the points raised by the reviewer and am happy to recommend publication